# Imidazoles and Quaternary Ammonium Compounds as Effective Therapies against (Multidrug-Resistant) Bacterial Wound Infections

**DOI:** 10.3390/antibiotics13100949

**Published:** 2024-10-10

**Authors:** Lauren Van de Vliet, Thijs Vackier, Karin Thevissen, David Decoster, Hans P. Steenackers

**Affiliations:** 1MiCA Lab, Centre of Microbial and Plant Genetics (CMPG), Department Microbial and Molecular Systems, KU Leuven, 3001 Leuven, Belgium; 2CMPG-PFI (Plant-Fungus Interactions Group of Centre of Microbial and Plant Genetics), Department Microbial and Molecular Systems, KU Leuven, 3001 Leuven, Belgium

**Keywords:** antifungal azoles, bacterial wound infections, antimicrobial resistance, synergistic combinations, experimental evolution

## Abstract

Background/Objectives: The rise and spread of antimicrobial resistance complicates the treatment of bacterial wound pathogens, further increasing the need for newer, effective therapies. Azoles such as miconazole have shown promise as antibacterial compounds; however, they are currently only used as antifungals. Previous research has shown that combining azoles with quaternary ammonium compounds yields synergistic activity against fungal pathogens, but the effect on bacterial pathogens has not been studied yet. Methods: In this study, the focus was on finding active synergistic combinations of imidazoles and quaternary ammonium compounds against (multidrug-resistant) bacterial pathogens through checkerboard assays. Experimental evolution in liquid culture was used to evaluate the possible emergence of resistance against the most active synergistic combination. Results: Several promising synergistic combinations were identified against an array of Gram-positive pathogens: miconazole/domiphen bromide, ketoconazole/domiphen bromide, clotrimazole/domiphen bromide, fluconazole/domiphen bromide and miconazole/benzalkonium chloride. Especially, miconazole with domiphen bromide exhibits potential, as it has activity at a low concentration against a broad range of pathogens and shows an absence of strong resistance development over 11 cycles of evolution. Conclusions: This study provides valuable insight into the possible combinations of imidazoles and quaternary ammonium compounds that could be repurposed for (topical) wound treatment. Miconazole with domiphen bromide shows the highest application potential as a possible future wound therapy. However, further research is needed into the mode of action of these compounds and their efficacy and toxicity in vivo.

## 1. Introduction

In Europe, an estimated 1.5–2 million people suffer from acute or chronic wounds. The burden of these types of wounds on the healthcare system cannot be underestimated but is often overlooked. As millions of patients are impacted, wounds are often described as ‘A Silent Epidemic’ [1]. Contributing to this issue is the possibility of infection, which delays wound healing [2]. Wound infections are prevalent: one out of three nosocomial infections are wound infections. These infections lead to a reduced quality of life, extended hospital stays and even contribute to 70–80% mortality of all nosocomial infections [3,4].

Wound infection can occur rapidly, as the surrounding tissue of the patient is colonized by microorganisms from their own flora or from the environment [5]. When the surface of the skin is breached, the exposed subcutaneous layer becomes a suitable environment for the colonization and proliferation of (pathogenic) microorganisms [6]. The most frequently isolated species from infected tissues are the Gram-negative *Pseudomonas aeruginosa*, *Escherichia coli*, *Klebsiella pneumoniae* and *Acinetobacter baumannii* and the Gram-positive *Staphylococcus aureus*, *Staphylococcus haemolyticus*, *Enterococcus faecalis* and *Staphylococcus epidermidis* [2,4,7,8,9,10]. These pathogens can colonize tissues in a monomicrobial or polymicrobial infection [4]. In chronic wounds, polymicrobial infections appear more often, where bacteria form a biofilm that attaches to the tissue [11].

Once the tissue has been infected, proper treatment regimens need to be applied to prevent the pathogens from migrating to deeper tissues and causing secondary infection [9]. These treatment regimens are, however, strongly impacted by the rapid increase in and spread of antimicrobial resistance [8]. Frequently, antimicrobial resistance is detected in wound pathogens [4,12,13,14]. The existence of multidrug-resistant strains makes topical and systemic antibiotic treatment difficult, which is often crucial in treating wound infections in combination with debridement and cleansing strategies [11,15]. Moreover, topical antibiotics such as fusidic acid and mupirocin have even been restricted in their use, as this could further promote resistance [16].

With the rising threat of antimicrobial resistance, more effective therapies are highly needed. Alternatives proposed for antibiotics are antimicrobial peptides [17,18], essential oils [19,20], honey [21,22] and nanoparticle treatments [8]. In addition to looking for alternative strategies, existing antimicrobials can also be repurposed to a different application field. The advantage of repurposing already commercialized drugs is that the pipeline to market approval is reduced, as certain important parameters, such as the pharmacological characteristics, safety and toxicology evaluations, are already documented. As a result, phase I clinical trial studies for safety would not need to be conducted if data from previous safety studies can be used [23,24].

A promising drug class that can be reused is the class of antifungal azoles, such as miconazole, clotrimazole and ketoconazole. These abovementioned compounds are part of the imidazole class, which is the largest class of antifungal drugs, along with triazole antifungals [25]. Imidazoles are characterized by two nitrogen atoms in the azole ring [26], while triazoles contain three nitrogen atoms [27]. Imidazoles such as miconazole are characterized by fungicidal activity against planktonic cultures of *Candida* species, with limited activity against *Candida* biofilm cultures. To increase the activity of miconazole against fungal biofilms, Tits et al. (2020) screened for potentiators of miconazole’s antibiofilm activity. They found that the quaternary ammonium compound (QAC) domiphen bromide is a potentiator for miconazole fungicidal activity, leading to a significant reduction in *Candida* cell counts in treated planktonic or biofilm cultures as compared to monotherapy. Remarkably, this reduction was found for both sensitive *C. albicans* and fluconazole-resistant *C. albicans* isolates and azole-resistant *C. glabrata* and *C. auris* [28].

In addition to their antifungal activity, these imidazoles are also characterized by antibacterial activity against (drug-resistant) Gram-positive bacteria [16,29], including common wound pathogens, such as *S. aureus*, *Streptococcus* and *Enterococcus.* The activity of the imidazoles is, however, limited against Gram-negatives [30]. Similar to miconazole, QACs such as domiphen bromide are used in healthcare. These compounds are applied as antiseptics with broad-spectrum antimicrobial activity [31], including antifungal, antiviral and antibacterial activity, although Gram-negative bacteria are often less susceptible [32]. Commonly used QACs, such as benzalkonium chloride or octenidine dihydrichloride, can be used directly on the skin or integrated in wound dressings [33,34].

There is, however, no information available about the activity of the combination of miconazole and domiphen bromide against bacterial pathogens. The current study therefore investigates the effect of combining imidazoles and QACs against bacterial (wound) pathogens. In studying their activity against a range of pathogens, this study will provide new insights on alternative combinations of antimicrobials, which can then be further validated in more relevant in vivo set-ups.

## 2. Results

### 2.1. The Combination of Miconazole with Domiphen Bromide Is Active against (Resistant) Gram-Positive Pathogens

Firstly, the effect of miconazole and domiphen bromide against four bacterial pathogens frequently isolated from wound infections was assessed: the Gram-negative *A. baumannii* RD5SR3 and *P. aeruginosa* PA14 and the Gram-positive *S. aureus* SH1000 and *S. epidermidis* SE40 (clinical isolate). The minimum inhibitory concentration (MIC) and fractional inhibitory concentration (FIC) indices of the mono- and combination therapy are presented in Table 1. Synergistic activity (FIC ≤ 0.5) of miconazole with domiphen bromide was found for all the tested strains, except for *P. aeruginosa.* The MICs of miconazole in monotherapy against *A. baumannii* and *P. aeruginosa* exceeded the highest included concentration (>399 and 408 µg/mL), which was not the case for *S. aureus* (19.5 µg/mL) and *S. epidermidis* (14.6 µg/mL).

As miconazole with domiphen bromide proved to be a promising synergistic combination against Gram-positive strains, this combination was further tested against a broader set of Gram-positive pathogens. Two additional clinical isolates were included, *S. aureus* MRSA and *S. epidermidis* CL7, along with the commonly isolated wound pathogens *E. faecium*, *S. pyogenes* and *S. dysgalactiae* and two other pathogens from the order *Bacillales*, *B. cereus* and *L. monocytogenes.* The obtained MICs of the planktonic cultures and calculated FICs are included in Table 1. Against *S. aureus* MRSA, *S. epidermidis* CL7, *S. dysgalactiae*, *B. cereus* and *L. monocytogenes*, synergistic antibacterial activity was found (FIC ≤ 0.5). Additivity (0.5 < FIC ≤ 1.0) was observed for the combination therapy against *E. faecium* and *S. pyogenes.* For the methicillin-resistant isolate *S. aureus* MRSA, a higher average MIC (127.4 µg/mL) of miconazole was found than for the lab-strain *S. aureus* SH1000 (19.5 µg/mL). Similarly, miconazole showed a higher MIC (108 µg/mL) against *E. faecium* than against *S. aureus* SH1000. This higher MIC was not seen with the clinical isolate *S. epidermidis* CL7 (4.9 µg/mL), which is resistant to multiple antibiotics, and *B. cereus* and *L. monocytogenes.* Additionally, the MICs of miconazole against the two microaerobic strains (*S. dysgalactiae* and *S. pyogenes*) are similar or lower than those against the aerobic strains, indicating the miconazole activity was retained in a microaerobic environment. Hence, miconazole and domiphen bromide showed promise as a synergistic antibacterial combination against seven of the nine tested Gram-positive pathogens. Next, the activity of domiphen bromide in combination with other imidazoles and triazoles was investigated.

### 2.2. Ketoconazole and Fluconazole Combined with Domiphen Bromide Show Activity against Sensitive and Multidrug-Resistant Staphylococci

Two additional imidazoles (clotrimazole and ketoconazole) and two triazoles (voriconazole and fluconazole) were included and tested against the lab-strain *S. aureus* SH1000 and the clinical isolate *S. epidermidis* CL7 (multidrug-resistant strain, see M and M). One lab-strain and one multidrug-resistant isolate were chosen to evaluate whether the clinical isolate showed different susceptibility to different combination therapies compared to the lab strain. The MICs and FICs of these compounds are presented in Table 2. Out of the tested combinations, the combinations based on domiphen bromide with either ketoconazole or fluconazole displayed synergistic antibacterial activity against *S. aureus* SH1000 and *S. epidermidis* CL7. Clotrimazole with domiphen bromide showed synergistic antibacterial activity against *S. aureus* SH1000, while no synergistic activity was found against *S. epidermidis* CL7. The second triazole, voriconazole, with domiphen bromide showed additive antibacterial activity against both bacteria.

The MICs of the triazoles and ketoconazole in monotherapy were consistently higher (>174.7 µg/mL for voriconazole, >153.1 µg/mL for fluconazole and >265.8 µg/mL for ketoconazole) than the MICs of miconazole. As a next step, different combinations based on miconazole and various QACs were tested.

### 2.3. Miconazole with Benzalkonium Chloride Has Synergistic Activity against Sensitive S. aureus and Additive Activity against Drug-Resistant S. epidermidis

Four additional quaternary ammonium compounds (QACs) were tested, with three mono-QACs (benzethonium chloride, cetrimonium chloride and benzalkonium chloride) and one bis-QAC (octenidine dihydrochloride). All compounds were tested with miconazole against the lab-strain *S. aureus* SH1000 and the clinical isolate *S. epidermidis* CL7. The results are depicted in Table 3. Only miconazole with benzalkonium chloride showed synergistic antibacterial activity against *S. aureus* SH1000. No synergistic activity of this combination was found against *S. epidermidis* CL7. The other included QACs displayed additive antibacterial activity.

### 2.4. Miconazole and Domiphen Bromide in Combination Therapy Are Evolutionarily Robust during Experimental Evolution 

From all included combinations, miconazole with domiphen bromide showed the most promising synergistic activity against Gram-positive pathogens. To evaluate possible resistance development against the compounds, experimental evolution with *S. aureus* SH1000 treated with a monotherapy of miconazole, domiphen bromide and a 1:1 combination therapy of miconazole and domiphen bromide was performed. Additionally, a conventional antibiotic that is often used in wound care, fusidic acid, was included as a benchmark. The evolution of the MICs of the four *S. aureus* lineages of each therapy are depicted in Figure 1. Over the course of 11 cycles, no increase in the MIC was detected for the monotherapy of miconazole. A reduction in the MIC of miconazole was detected instead: in three lineages, a 2-fold decrease in the MIC was found, while in one lineage, a √2-fold decrease was observed. With domiphen bromide, a √2-fold to 2-fold increase was seen in cycle 10 for the three populations. In the combination therapy, a √2-fold increase in the MIC was detected for domiphen bromide and miconazole in two lineages, while no increase was found in the two other lineages after 11 cycles.

In contrast, under fusidic acid therapy, a large increase in the MIC rapidly emerged: after cycle 3, a 16-fold increase was found for all four lineages (1/2× MIC = 0.008 µg/mL to > 8× MIC, which is higher than 0.126 µg/mL), as all lineages overgrew the entire included concentration range. The concentration range was then expanded to 64× MIC − 1/2√2× MIC. After cycle 4, the increase continued with three of the included lineages having an MIC of 64× MIC and one displaying an MIC exceeding the concentration range. Again, the concentration range was increased to 512× MIC − 2√2× MIC. One lineage displayed an MIC higher than 512× MIC in the subsequent cycle (cycle 5), so the concentration range was expanded a final time to 4096× MIC − 16√2× MIC. After cycle 7, an increase in the MIC was again detected for the four lineages. Starting from cycle 8, the MICs for the four lineages stabilized, ending at 32.3 µg/mL (1024√2× MIC), two times 45.7 µg/mL (2048× MIC) and 22.8 µg/mL (1024× MIC). Compared to miconazole and domiphen bromide in mono- and combination therapy, a rapid increase in resistance development was found over the course of 11 cycles. Initially, the MIC of *S. aureus* SH1000 for fusidic acid (0.016 µg/mL) was also notably lower than the MIC for the azoles or QACs included in this research. After cycle 4, all lineages exceeded the clinical breakpoint for the systemic treatment of fusidic acid against *Staphylococci*, which is 1 µg/mL (Figure 1c, indicated in red) [35,36].

**Figure 1 antibiotics-13-00949-f001:**
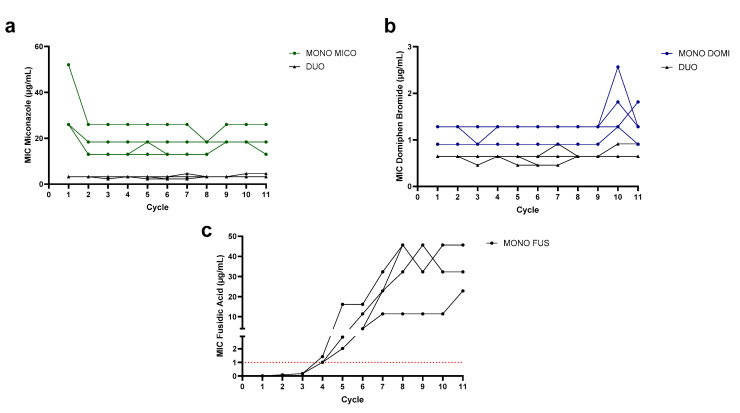
Evolution of the MIC (μg/mL) of (**a**) miconazole in monotherapy (green) and combination therapy (black), (**b**) domiphen bromide in monotherapy (blue) and combination therapy (black) and (**c**) fusidic acid over the course of 11 cycles (each 18 h) of the evolution experiment. Indicated in (**c**) in the red dotted line is the clinical breakpoint for fusidic acid (1 µg/mL), as determined by EUCAST for *Staphylococci* [36].

After experimental evolution, the possible increase in resistance in the evolved populations was validated using a spot assay with the mono- and combination therapy. The spot assay allowed us to determine the MICs in a more structured environment. The ratio of the zones of inhibition of the evolved populations compared to the ancestors are depicted in Figure 2. Interestingly, the ZOI of the combination therapy was similar to or larger than the ZOI of the monotherapies for all evolved populations, while lower absolute concentrations of the compounds were spotted in the combination therapy. Miconazole in duotherapy had an inhibition zone at 26 µg/mL, while in monotherapy, it only showed a similar inhibition zone at 208 µg/mL. Similarly, domiphen bromide in duotherapy showed inhibition at 5.2 µg/mL, while in monotherapy, at 58 µg/mL. The synergistic effect is thus maintained in a more structured environment, with activity of the compounds at concentrations ten times lower than the monotherapy. In the miconazole monotherapy spot assay (Figure 2a), a 0.2 to 0.3 decrease in the inner ZOI (light gray bars) was found for all four miconazole-evolved populations compared to the ancestors. Also, for the domiphen bromide monotherapy spot assay (Figure 2b), three out of four domiphen-evolved populations showed a 0.3 to 0.35 decrease in the ZOI, indicating a reduced susceptibility to the compound. This contrasted with our findings in the evolution experiment, where only one lineage showed an increase in the MIC. One combination therapy-evolved population (DUO EVO 4) also showed a decrease in the ZOI in this domiphen bromide monotherapy spot assay. This population corresponds to the population which showed an increase at cycle 11 (see Figure 1b). When the combination therapy was spotted, no strong decreases in ZOIs were found in the evolved populations compared to the ancestral populations. This could indicate that the resistance development against the combination therapy is limited, further confirming what was found in the evolution experiment.

## 3. Discussion

The rise in antimicrobial resistance requires modern medicine to reassess the (over)usage of conventional antibiotics, as pathogens are becoming more difficult to treat [37]. This increase in antibiotic resistance is also found for bacterial wound pathogens: previous analyses of wound samples report an increase in antibiotic resistance in prevalent clinical isolates. Around 29% of these strains were multidrug-resistant, with *S. aureus* and *P. aeruginosa* being the most frequently detected [3,38]. This multidrug-resistant character was also seen in this paper’s included clinical isolates (*S. aureus* MRSA, *S. epidermidis* CL7/SE40) used in our checkerboard assays, although these isolates did not originate from wounds. Antibiotic resistance of *S. aureus* especially is frequently reported against topical antimicrobials such as fusidic acid, which is often prescribed to treat vancomycin- or methicillin-resistant *S. aureus* [39]. In addition to antibiotic resistance, reduced sensitivity against antiseptics such as chlorhexidine or QACs is documented due to multidrug efflux mechanisms [40,41].

Despite this widespread resistance, wound care is still reliant on the use of antibiotics and antiseptics to prevent (systemic) infection [40,42,43]. More resistance-proof, effective therapies are thus needed. Miconazole is proposed as an alternative, as it has been proven to even have activity against methicillin- and fusidic acid-resistant *Staphylococcus* strains [29]. This drug was introduced on the market over 40 years ago, mostly used as a broad-spectrum topical antifungal [44]. In this study, the activity of different combinations of antifungal azoles with quaternary ammonium compounds against prevalent wound pathogens was assessed to explore possible new strategies of repurposing these drugs. 

Firstly, synergistic antibacterial activity of miconazole with domiphen bromide was discovered against one Gram-negative pathogen and against seven out of the nine tested Gram-positive bacteria, including three multidrug-resistant strains. Its synergistic activity is thus not pathogen-specific. This promising combination led to a strong reduction in the active concentration of both the imidazole and the quaternary ammonium compound (reduction in MIC_90_ of miconazole > 8-fold, in domiphen bromide > 3-fold). A reduction in the active concentration, as both miconazole and domiphen bromide could be administered at a low concentration, could further minimize the risk for contact allergy to either the antifungal or quaternary ammonium compound [45,46]. However, the MIC of miconazole against the Gram-negative bacteria remained high (~>400 µg/mL), leading to reduced applicability against these pathogens. While this combination showed synergistic activity against multiple (drug-resistant) *Staphylococcus* strains, *S. dysgalactiae*, *B. cereus* and *L. monocytogenes*, additivity was found against *E. faecium* and *S. pyogenes.*

Azoles such as miconazole exert their antifungal activity through the inhibition of lanosterol 14 alpha-demethylase (CYP450 enzyme), which impedes ergosterol biosynthesis and leads to growth inhibition. Additionally, its antifungal activity has been attributed to the generation of reactive oxygen species (ROS) [47,48] and the production of K^+^ efflux through the disruption of the membrane potential [49]. Homologues of these 14 alpha-demethylases are hypothesized to be present in *Staphylococcus* [29], which could explain miconazole’s antibacterial activity, although these homologues were only experimentally found in a select subset of bacteria such as *Mycobacterium* [50]. Similar to this ability, miconazole can bind to flavohemoglobin, which increases ROS production [16,29,51,52]. Combining this antibacterial activity with domiphen bromide has not yet been tested against bacteria. In fungi, domiphen bromide works as a potentiator by increasing the uptake of miconazole and through the release of sequestered miconazole, resulting in higher reactive oxygen stress [48]. Similarly, we expect domiphen bromide to increase the uptake of miconazole. The mode of action of QACs such as domiphen bromide includes the disruption of the bacterial membrane through the ionic interaction of the positively charged N-atom with the negatively charged bacterial membrane. After binding, the side chain of the QAC interacts with the phospholipid layer, disrupting its structure and leading to permeabilization and cell leakage [53]. Domiphen bromide’s ability to disrupt the outer membrane has already been shown in Gram-negatives, which led to a synergistic combination therapy with the antibiotic colistin [54].

A promising second combination identified from the screening is ketoconazole with domiphen bromide, which has synergistic antibacterial activity against *S. aureus* SH1000 and *S. epidermidis* CL7. Ketoconazole activity against *Staphylococci* has been described as less effective as miconazole [55], which was also seen in the elevated MIC for both tested strains in our study. The mode of action of this imidazole is similar to miconazole, through the inhibition of ergosterol synthesis [56]. Like miconazole, ketoconazole can induce K^+^ efflux in fungi. However, its effect on efflux is smaller than for miconazole, which could explain the lower antibacterial activity of this compound [49]. Additionally, it has also been shown that ketoconazole can act as a potentiator for other antibiotics, inhibiting the expression of efflux pump genes [55]. We hypothesize that the synergistic activity in our assays could be due to a similar mechanism as found with miconazole and domiphen bromide, where domiphen bromide increases the uptake of ketoconazole. Combined with ketoconazole’s ability to inhibit efflux, this could lead to a strong intracellular accumulation of the compound. Additional research will be needed into the mechanism underlying this synergistic activity, which can be performed through compound localization of a fluorescent imidazole analogue, ROS dyes or through omics profiling [48,57].

Furthermore, synergistic activity was found for clotrimazole and domiphen bromide against *S. aureus* SH1000 and additivity against *S. epidermidis* CL7. Clotrimazole has been previously shown to have antibacterial activity [16]. Synergy with domiphen bromide against *C. albicans* biofilms was also found before, possibly through the same mode of action as miconazole [28]. Possible differences in synergy between SH1000 and CL7 could be attributed to differences in the inherent antibiotic resistance mechanisms, although no collateral sensitivity or cross-resistance has been documented yet between antibiotics and azoles.

Both *Staphylococci* seemed to be resistant to the triazoles, as found earlier in the literature [52,58]. While these pathogens had a higher MIC against both components, fluconazole with domiphen bromide had synergistic activity against the *Staphylococci.* Voriconazole with domiphen bromide was found to have additive activity. Triazoles have a similar mode of action to imidazoles through the inhibition of 14-alpha-demethylase [59], although their ability to bind to bacterial flavohemoglobins is different. It has been previously reported that the presence of bulky side chains, geometric orientation of the azole ring and interactions with N-substituents can influence the flavohemoglobin binding affinity of the azoles. Triazoles show the lowest affinity due to their bulky side chains, while the presence of an extra Cl-atom in miconazole leads to a higher affinity [60,61]. If the affinity of the imidazole/triazole is insufficient to bind to flavohemoglobin and thus increase ROS production, increased permeabilization of the membrane by domiphen bromide is not expected to increase the activity of the combination therapy, as its affinity for the flavohemoglobin remains unchanged. This could explain the presence of synergy with fluconazole and additivity with voriconazole. Voriconazole is a chemical derivative of fluconazole, modified with a fluoropyrimidine ring, which could influence its electronic interactions with flavohemoglobin [62].

A final promising combination is miconazole with benzalkonium chloride. This synergistic activity was only found with *S. aureus* SH1000. Synergy between miconazole and the QAC chlorhexidine was also found against *S. aureus* MRSA [29]. For the mode of action of different QACs, the efficacy of the disruption of the bilayer depends on the length of the alkyl side chain and the composition of the bacterial membrane [63,64,65]. This difference in permeabilization activity could explain why miconazole with benzalkonium chloride showed synergistic activity against *S. aureus* SH1000 and not against *S. epidermidis* CL7. The two strains most likely differ in membrane composition, which could be due to strain differences or due to their antibiotic resistance profile. In turn, this could influence benzalkonium chloride’s activity to disrupt the differently structured membrane. Similarly, the difference in the alkyl side chain between the different QACs could influence their permeabilization activity, leading to less effective disruption of the membrane and thus no potentiation of the antimicrobial activity of miconazole.

Although the found duotherapies with synergistic activity are promising, the usage of QACs for antibacterial purposes should be approached with caution. QAC and antiseptic resistance is frequently found in bacterial species (e.g., *S. aureus*), which can also lead to cross-resistance and the evolution of resistance against commonly used antibiotics [66,67]. To this end, possible resistance development of one of the promising duotherapies, miconazole with domiphen bromide, was evaluated.

No increase was found in the MIC of miconazole-treated populations over the course of 11 days. Similarly, no strong increase was found in the outer ZOI for miconazole in the spot assay. For domiphen bromide, no increase in the MIC was found in the evolution experiment, except for in one lineage in the mono- and combination therapy with a √2-fold increase. However, in the spot assay, an increased resistance was seen in three domiphen-evolved populations. This indicates that resistance development against the QAC can emerge over short time frames (11 cycles) and that this resistance mechanism has a stronger effect in a more structured environment (agar) than liquid medium. In contrast, the combination-treated populations seemed to be more evolutionarily robust in the evolution experiment or spot assay.

Resistance development of bacteria against miconazole has not yet been studied, as this drug is not used against bacterial pathogens. Against fungi, no resistance has been documented, despite the widespread usage of miconazole. The lack of resistance development could be attributed to its multiple modes of action or to the sole topical use of miconazole [44,68,69,70]. Resistance development against domiphen bromide specifically is not documented; however, as broad-spectrum resistance mechanisms exist against QACs, they could also lead to resistance against domiphen bromide [71].

For the benchmark antibiotic fusidic acid, a rapid and strong increase was found over 11 days in the MIC for *S. aureus* SH1000, while the bacterium initially showed high susceptibility to the antibiotic. This conventional antibiotic is often used to clear topical *S. aureus* skin infections. Increased usage of this drug has led to the emergence of resistant *S. aureus* isolates and to outbreaks [39,72]. Fusidic acid resistance is often marked by a high resistance frequency, and can lead in certain cases to a high and rapid increase (>2048-fold) in the MIC [73].

Compared to this conventional antibiotic, miconazole and domiphen bromide showed promising evolutionarily robust potential in combination therapy. Previously, this lack of resistance development was also found with a combination of an antibiotic (ciprofloxacin) and a QAC (dequalinium chloride), which supports the resistance-proof character of these types of combination therapies [74]. In general, duotherapies show great potential as tools against resistance development. However, the effect of drug interactions on resistance development in the literature yields conflicting evidence [75]. On the one hand, synergistic combinations can combat resistant bacteria and prevent resistance emergence, as the probability of two independent resistance mechanisms to emerge is much smaller than one resistance mechanism [76,77,78]. On the other hand, if these events are not independent, for example, in the case of multidrug resistance, synergistic combinations might not confer this advantage. Moreover, if the synergistic effect disappears due to resistance to one drug, the fitness of mutants resistant to one of the drugs is higher than the susceptible population, which could increase resistance development. This is, however, only the case around MIC levels, while therapeutic concentrations are often much higher [76]. Finally, Jahn et al. (2021) show that the effect on resistance evolution is largely combination-specific and depends more on whether developing resistance against the combination therapy requires a new genetic adaptation compared to the monotherapy [75]. Our assays can further support the first hypothesis, as resistance development in the monotherapy of the QAC emerged while no strong resistance emerged against the combination therapy. However, the experiment can still be prolonged to see if resistance still emerges against the combination therapy. Additionally, to evaluate resistance development in a more relevant setting, an evolution experiment can be performed with the compounds at clinically therapeutic concentrations, followed by serial passage of the surviving populations.

## 4. Materials and Methods 

### 4.1. Used Chemicals and Bacterial Strains

Stock solutions (100×) of the imidazoles (±)-miconazole nitrate salt (Sigma-Aldrich, St. Louis, MO, USA, M3512), ketoconazole (TCI Europe NV, Zwijndrecht, Belgium, K0045), clotrimazole (TCI Europe NV, C2867), and the triazoles fluconazole (TCI Europe NV, F0677) and voriconazole (TCI Europe NV, V0116) were prepared at 200 mM and the quaternary ammonium compounds (QACs), domiphen bromide (Sigma-Aldrich, 247480) and octenidine dihydrochloride (TCI Europe NV, O0388), were prepared at 20 mM. All abovementioned solutions were made in dimethyl sulfoxide (DMSO, VWR International, Radnor, PA, USA). Similarly, stocks (100×) of benzalkonium chloride (Sigma-Aldrich, 12060), benzethonium chloride (TCI Europe NV, B0044) and cetrimonium chloride (hexadecyltrimethylammonium chloride, Thermo Scientific Chemicals, Waltham, MA, USA, 10762402) were dissolved in dH_2_O at a concentration of 20 mM. 

*Acinetobacter baumanni* RD5SR3, *Pseudomonas aeruginosa* PA14, *Staphylococcus aureus* SH1000, *Bacillus cereus* ATCC7004, *Listeria monocytogenes* ScottA, *Enterococcus faecium* LMG 8148, *Streptococcus pyogenes* ATCC 12358, *Streptococcus dysgalactiae* subsp. *equisimilis* ATCC 12449 and two clinical isolates of *Staphylococcus epidermidis* strains, SE40 and CL7, and one clinical isolate of a methicillin-resistant *S. aureus* (MRSA n°34) were used for antimicrobial testing. *S. epidermidis* SE40 was isolated from a patient with bacteremia, *S. epidermidis* CL7 was isolated from a patient with osteomyelitis while the MRSA strain was isolated from a patient’s catheter at University Hospital Leuven (UZ Leuven). Antibiograms were performed at UZ Leuven on *S. epidermidis* SE40 and CL7, as recommended by the European Committee for Antibiotic Susceptibility Testing (EUCAST). *S. epidermidis* SE40 is resistant against oxacillin, levofloxacin, moxifloxacin, fusidic acid and trimethoprim, while *S. epidermidis* CL7 is resistant against oxacillin, levofloxacin, gentamicin, tobramycin, erythromycin, clindamycin, rifampicin, minocycline and co-trimoxazol.

Before each checkerboard assay, overnight cultures (ONCs) were prepared in either 5 mL lysogeny broth (LB; 10% Tryptone, 10% NaCl, 5% Yeast Extract) or 5 mL brain heart infusion broth (BHI, Becton Dickinson, Franklin Lakes, NJ, USA). BHI was used for microaerobic cultures of *Streptococcus pyogenes* and *Streptococcus dysgalactiae*, while LB was used for aerobic cultures of the other strains. To create a microaerobic environment, the GasPak™ EZ container system (Becton Dickinson) was used.

In the checkerboard assays (described below), the used growth medium for *A. baumanni*, *P. aeruginosa*, *S. aureus* SH1000, *S. epidermidis* SE40 and *S. aureus* MRSA was tryptic soy broth (TSB, Becton Dickinson). LB was used for *B. cereus*, *L. monocytogenes*, *E. faecium* and *S. epidermidis* CL7, while BHI in microaerobic conditions was used for *S. pyogenes* and *S. dysgalactiae*. TSB was also used as the growth medium for the experimental evolution set-up (described below). After incubation in overnight cultures, the bacterial cultures were each diluted 1:100 in the appropriate growth medium, which is used as an inoculum for the checkerboard assays.

### 4.2. Checkerboard Assays

To screen for different synergistic combinations between the imidazoles and QACs, checkerboard assays were performed as described in the previous literature [79,80]. Preparation of two-fold serial dilution series of the 100× stock of imidazoles in DMSO or QAC in DMSO or dH_2_O was followed by 1:10 dilution of the series in growth medium. Of these compound solutions, 20 µL was added to 100 µL bacterial inoculum (see used chemicals and bacterial strains) and 80 µL growth medium, resulting in an end volume of 200 µL. This resulted in a concentration range of 250–0.25 µM of the imidazole and 25–0.4 µM of the QAC. If a particular bacterial strain showed (no) growth across the entire concentration range, the concentration range was adjusted. The exact concentration ranges per bacterium can be found in the Appendix A. Monotherapies of the compounds were included at the same concentration ranges, as well as two sterile medium controls and untreated growth controls per checkerboard. The plates were then incubated for 24 h at 37 °C in aerobic conditions, except for *S. pyogenes* and *S. dysgalactiae*, which were incubated in microaerobic conditions using the GasPak™ EZ container system.

After incubation, the absorbance of the cultures (OD_595nm_) was measured using the Synergy Mx multimode reader (Agilent Technologies, Santa Clara, CA, USA), and the minimal inhibitory concentration (MIC) of the monotherapy and the combination therapy was determined, using 90% growth inhibition (MIC_90_) of the untreated control as a cutoff for the MIC [81]. If the bacterial growth exceeded the included concentration range, the highest concentration ×2 was included as the presumed MIC. Based on these MICs, the fractional inhibitory concentration (FIC) index could be calculated to identify synergistic combinations. The FIC values were calculated with the equation ∑FICI = FIC_A_ + FIC_B_ = (C_A_/MIC_A_) + (C_B_/MIC_B_) where MIC_A_ and MIC_B_ are the MICs of compound A and B alone, respectively, and C_A_ and C_B_ are the concentrations of the compounds in combination, respectively [28,80,81,82]. For each checkerboard assay, the minimal FICI was determined for the different concentrations that were ≥ MIC and reported per replicate. Combinations were considered as synergistic if FIC ≤ 0.5, additive if 0.5 < FIC ≤ 1.0, indifferent if 1.0 < FIC < 4.0 and antagonistic if FIC ≥ 4.0 [80,83,84]. At least 3 biological repeats were collected per combination and organism.

### 4.3. Experimental Evolution of Monotherapy vs. Combination Therapy

To assess the risk of the resistance development of the synergistic combination therapy, the resistance development of a monotherapy of fusidic acid, miconazole and domiphen bromide was compared against the resistance development of miconazole with domiphen bromide. Experimental evolution was used, as previously described in the literature, to evaluate an increase in the MIC over time [75,76]. Four ONCs of *S. aureus* SH1000 were density normalized to an inoculum of ~4 × 10^7^ CFU/mL in PBS 1× (8% NaCl, 0.2% KCl, 1.44% Na_2_HPO_4_, 0.24% of KH_2_PO_4_). This normalized culture was then diluted 1:10 in PBS 1x. For the monotherapies, a dilution series was made of 100× fusidic acid, miconazole and domiphen bromide stocks in DMSO, while for the combination therapy, a 100× 1:1 solution of miconazole and domiphen bromide was made and serially diluted. A range was included from 8× MIC to 1/(2√2)× MIC of the monotherapies and combination therapy, each step increasing with a factor of √2. These MIC values for miconazole and domiphen bromide were previously obtained from the checkerboard assays. For fusidic acid, the MIC was previously determined using a standard microbroth dilution method in Deepwell plates [85]. If the MIC surpassed the highest concentration, the range was expanded. These plates were stored at −20 °C between cycles. Each cycle, each well of the mother plate was diluted 1:100 in a 96-well Deepwell plate (ThermoFisher) with 490 µL 2× TSB and 500 µL *S. aureus* inoculum in PBS 1×.

Four replicates (lineages) each were included for the monotherapy of miconazole, the monotherapy of domiphen bromide and the combination therapy of the two components. Additionally, four control-evolved strains (strains evolved in pure growth medium with 1% DMSO) per treatment and sterile controls (TSB + 1% DMSO) were included. The cultures were then incubated for 18 h at 37 °C, while being shaken at 750 rpm.

After each incubation cycle, 100 µL of each well was stored at −80 °C, using 50% glycerol. Another 100 µL was used for absorbance measurements at 595 nm using the Synergy Mx multimode reader. From each replicate (each lineage), the population at the highest concentration where there was at least 40% growth of the control-evolved lineages was selected as an inoculum for the next cycle [76]. MIC determination from the absorbance measurements was performed as described for the checkerboard assays. All control-evolved lineages were transferred to the next cycle. Each population was then washed and density normalized in PBS 1×, followed by the same steps described as above. This process was repeated for 11 incubation cycles in total.

### 4.4. Spot Assay

To validate if the populations of the evolution experiment showed possible resistance development, a spot assay was performed, as described by Tits et al. (2020) [28]. Bacterial lawns of the ancestral, control-evolved and evolved populations were made through density normalization to OD_595nm_ = 0.55 and 1/20 dilution in PBS 1×, followed by bead plating of 100 µL. End concentrations of 500 µM of miconazole, 140 µM of domiphen bromide in monotherapy and 62.4 µM of miconazole and 12.5 µM of domiphen bromide in combination therapy were spotted. Quantification of the inhibition zone area was performed using the ImageJ 1.54f (Fiji) software, where the zone of inhibition (ZOI) (in cm^2^) was selected using the oval and measure function [86]. When partial growth was found within the ZOI, two zones were determined: one with total clearance (inner ZOI) and one with partial growth (outer ZOI). To compare with the ancestral ZOI, a ratio was calculated of each control-evolved and evolved population to their respective ancestor.

### 4.5. Statistical Analysis

MIC values of the checkerboard assays in mono- and combination therapy were subjected to robust regression and outlier removal (ROUT) in Graphpad Prism 10.1.2 with Q = 5%, assuming normality of the data [87]. Repeats that were detected as outliers were removed from the analysis.

## 5. Conclusions

Through checkerboard assays, this study was able to prove that miconazole with domiphen bromide had synergistic antibacterial activity against one Gram-negative and multiple Gram-positive pathogens. This combination holds promise, as its synergistic activity is present against multidrug-resistant clinical isolates and was found to be an evolutionarily robust combination. In comparison to the rapid resistance development against the antibiotic fusidic acid, no strong resistance development was found for this combination therapy. Additionally, combining other antifungal azoles with QACs yielded synergistic antibacterial activity, with ketoconazole/domiphen bromide, clotrimazole/domiphen bromide, fluconazole/domiphen bromide and miconazole/benzalkonium chloride emerging as positive hits.

As the effects of the antifungal azoles are often multifaceted, it is challenging to identify the mechanistic explanation behind the synergistic interactions. Therefore, detailed research of the mode of action of synergistic combinations, based on compound localization or omics techniques, is still needed. Moreover, this study included only the anti-planktonic effects of the compounds. Evaluating the antibiofilm effect of these promising combinations on bacterial wound pathogens also remains necessary, as biofilms often contribute to chronic wound infections [88,89]. Further, in vivo (or ex vivo) validation of these duotherapies is still needed, for example, in a pig skin explant model [90]. Previous in vivo experiments in a vulvovaginal candidiasis rat model indicate no increased inflammation, although cytotoxicity testing of the included compounds using an in vitro keratinocyte model will be necessary before moving to in vivo studies [28,91]. While further research is needed, this study can already pave the way for new resistance-proof topical wound treatments.

## Figures and Tables

**Figure 2 antibiotics-13-00949-f002:**
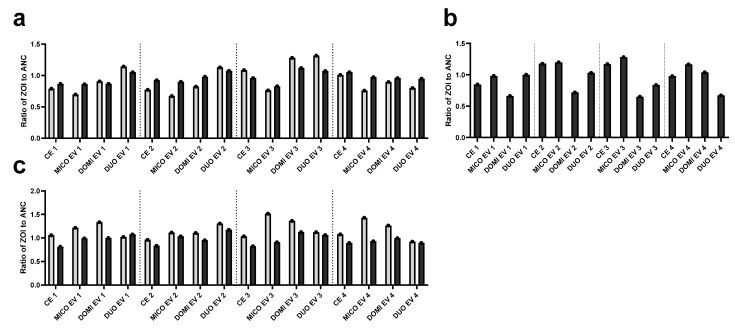
Ratio of the zone of inhibition (ZOI) of the evolved populations to the ZOI of their respective ancestral population obtained in the spot assay. The control-evolved (CE), miconazole-treated (MICO EV), domiphen bromide-treated (DOMI EV) and combination therapy (DUO EV) were obtained through experimental evolution. The populations were treated with (**a**) 208 µg/mL (8× MIC) miconazole, (**b**) 58 µg/mL (32 MIC) domiphen bromide and (**c**) 26 µg/mL (8× MIC) miconazole with 5.2 µg/mL (8× MIC) domiphen bromide. Both the zone of complete clearance (light gray) and the outer edges of the ZOI with partial growth (dark gray) were quantified using the ImageJ 1.54f (Fiji) analysis software.

**Table 1 antibiotics-13-00949-t001:** Overview of the average MIC values of miconazole (MICO), domiphen bromide (DOMI) for *A. baumannii* (*n* = 3), *P. aeruginosa* (*n* = 3), *S. aureus* SH1000 (*n* = 4), *S. epidermidis* SE40 (*n* = 4), *S. aureus* MRSA (*n* = 5), *S. epidermidis* CL7 (*n* = 4), *E. faecium* (*n* = 5), *S. pyogenes* (*n* = 4), *S. dysgalactiae* (*n* = 4), *B. cereus* (*n* = 4) and *L. monocytogenes* (*n* = 3) of the checkerboard analyses after 24 h of incubation. The cumulative fractional inhibitory concentration index (ΣFIC) was calculated based on the MICs of both components in monotherapy and combination therapy. At FIC ≤ 0.5, combinations are considered as synergistic (indicated in green) with the average FICI depicted in the table.

Strain		MIC Monotherapy		MIC Combination Therapy		FICI
		µM	µg/mL	µM	µg/mL	
*A. baumannii* RD5SR3	MICO	>960.0	>399.4	1.6	0.7	
	DOMI	12.5	5.2	6.3	2.6	0.5
*P. aeruginosa* PA14	MICO	>980.0	>407.8	5.5	2.3	
	DOMI	16.7	6.9	10.4	4.3	0.7
*S. aureus* SH1000	MICO	46.9	19.5	10.5	4.4	
	DOMI	12.5	5.2	3.9	1.6	0.5
*S. epidermidis* SE40	MICO	35.2	14.6	10.6	4.4	
	DOMI	6.3	2.6	1.2	0.5	0.5
*S. aureus* MRSA	MICO	306.3	127.4	25.1	10.4	
	DOMI	22.5	9.3	6.3	2.6	0.4
*S. epidermidis* CL7	MICO	11.7	4.9	1.7	0.7	
	DOMI	3.1	1.3	1.0	0.4	0.5
*E. faecium* LMG 8148	MICO	259.4	108.0	20.4	8.5	
	DOMI	17.5	7.3	8.8	3.6	0.7
*S. pyogenes* ATCC 12358	MICO	5.9	2.4	1.0	0.4	
	DOMI	2.0	0.8	0.8	0.3	0.8
*S. dysgalactiae* ATCC 10706	MICO	7.8	3.3	1.3	0.6	
	DOMI	2.7	1.1	0.8	0.3	0.5
*B. cereus* ATCC 7004	MICO	7.8	3.3	2.4	1.0	
	DOMI	9.4	3.9	1.8	0.7	0.5
*L. monocytogenes* ScottA	MICO	3.9	1.6	0.8	0.3	
	DOMI	6.3	2.6	1.6	0.6	0.5

**Table 2 antibiotics-13-00949-t002:** Overview of average MIC values of *S. aureus* SH1000 and *S. epidermidis* CL7 (CLOTRI = clotrimazole *n* = 6 for SH1000 *n* = 3 for CL7), KETO = ketoconazole (*n* = 6 for SH1000, *n* = 4 for CL7), VORI = voriconazole (*n* = 3), FLUCO = fluconazole (*n* = 3), DOMI = domiphen bromide) of the checkerboard analyses after 24 h of incubation. The cumulative fractional inhibitory concentration index (Σ FIC) was calculated based on the MICs of both components in monotherapy and combination therapy. At FIC ≤ 0.5, combinations are considered as synergistic (indicated in green) with the average FICI depicted in the table.

Strain		MIC Monotherapy		MIC Combination Therapy		FICI
		µM	µg/mL	µM	µg/mL	
*S. aureus* SH1000	CLOTRI	49.0	16.9	7.9	2.7	
	DOMI	13.5	5.6	4.2	1.7	0.5
	KETO	>500.0	>265.8	50.5	26.8	
	DOMI	10.9	4.5	3.1	1.3	0.4
	VORI	>1000.0	>349.3	1.7	0.6	
	DOMI	8.3	3.5	8.3	3.5	1.0
	FLUCO	>1000.0	>306.3	1.7	0.5	
	DOMI	34.9	14.5	5.2	2.2	0.5
*S. epidermidis* CL7	CLOTRI	52.1	18.0	15.6	5.4	
	DOMI	4.2	1.7	1.6	0.6	0.8
	KETO	>537.5	>285.6	134.4	71.4	
	DOMI	3.1	1.3	0.8	0.3	0.5
	VORI	>500.0	>174.7	0.5	0.2	
	DOMI	1.6	0.6	1.6	0.6	1.0
	FLUCO	>500.0	>153.1	1.1	0.3	
	DOMI	3.1	1.3	1.6	0.6	0.5

**Table 3 antibiotics-13-00949-t003:** Overview of average MIC values of *S. aureus* SH1000 and *S. epidermidis* CL7 of the checkerboard analyses (MICO = miconazole with BENZETH = benzethonium chloride (*n* = 3), CETRI = cetrimonium chloride (*n* = 3), BENZALK = benzalkonium chloride (*n* = 5 for SH1000, *n* = 3 for CL7), OCT = octenidine dihydrochloride (*n* = 3)) after 24 h of incubation. The cumulative fractional inhibitory concentration index (Σ FIC) was calculated based on the MICs of both components in monotherapy and combination therapy. At FIC ≤ 0.5, combinations are considered as synergistic (indicated in green) with the average FICI depicted in the table.

Strain		MIC Monotherapy		MIC Combination Therapy		FICI
		µM	µg/mL	µM	µg/mL	
*S. aureus* SH1000	MICO	31.3	13.0	4.5	1.9	
	BENZETH	8.3	3.7	4.2	1.9	0.6
	MICO	26.0	10.8	8.8	3.7	
	CETRI	7.3	2.7	2.6	0.9	0.8
	MICO	81.3	33.8	14.6	6.1	
	BENZALK	18.8	5.8	3.4	1.1	0.4
	MICO	20.8	8.7	4.6	1.9	
	OCT	2.1	1.3	0.9	0.6	0.6
*S. epidermidis* CL7	MICO	13.0	5.4	4.1	1.7	
	BENZETH	8.3	3.7	3.6	1.6	0.8
	MICO	18.3	7.6	4.6	1.9	
	CETRI	3.1	1.1	1.6	0.6	0.8
	MICO	13.0	5.4	3.1	1.3	
	BENZALK	16.7	5.2	7.3	2.3	0.6
	MICO	15.6	6.5	4.1	1.7	
	OCT	1.8	1.1	0.7	0.4	0.7

## Data Availability

The original contributions presented in the study are included in the article; further inquiries can be directed to the corresponding authors.

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
