# Peer review of "Imidazoles and Quaternary Ammonium Compounds as Effective Therapies against (Multidrug-Resistant) Bacterial Wound Infections"

_antibiotics, 2024, doi:10.3390/antibiotics13100949_

Round 1
Reviewer 1 Report
Comments and Suggestions for Authors
With the rise of multi-drug resistance of bacteria infections, novel and effective therapies are in high demand. Building on the synergistic antifungal activity of imidazoles and quaternary ammonium compounds (QAC) combinations, the authors investigated their acictivy against resistant bacterial pathogens and confirmed a similarly strong synergistic effect. This study highlights the in vitro potential of repurposing approved drugs to overcome the multi-drug resistence of bacterial pathogens.
Suggestions:
1) What is the difference of modes of action between miconazole and fusidic acid? Can combinations of fusidic acid and QACs overcome the bacterial resistence ?
2) The efficacy and synergistic effect should ideally be confirmed in vivo, but this is not a requirement in this work.
3) Although understanding the action mode of imidazoles and quaternary ammonium compounds combinations would be difficult, it is crucial to address potential resistance to QACs in future studies.
Author Response
We included the response in the attachment, thank you kindly for reviewing it.

Reviewer 2 Report
Comments and Suggestions for Authors
The article focuses on the search for new methods of treating microbial infections of wounds. The authors investigated antimicrobial compounds with already known pharmacological effects. To this end, they selected several compounds from the azole group, which exhibit synergistic antifungal activity with quaternary ammonium compounds. However, the antibacterial activity of such combinations has not yet been explored. The authors focused on finding active synergistic combinations of selected imidazoles with quaternary ammonium compounds against (multidrug-resistant) bacterial pathogens. They identified several promising combinations against Gram-positive pathogens: miconazole/domiphen bromide, ketoconazole/domiphen bromide, clotrimazole/domiphen bromide, fluconazole/domiphen bromide, and miconazole/benzalkonium chloride. Especially, miconazole with domiphen bromide exhibits potential, as it shows activity at low concentrations against a broad range of pathogens and does not induce resistance development over 11 cycles of evolution. The choice of studied compounds is interesting due to their diverse structural composition, and the possibility of repurposing existing drugs with known pharmacological parameters is an additional advantage.
The search for new antimicrobial therapies is currently one of the most topical subjects. The rise in antibiotic-resistant pathogenic bacteria poses a serious threat to public health. Despite the urgent need to develop new antibiotics, bacterial resistance to these compounds is growing faster than their market introduction. Developing solutions that do not generate resistance in microorganisms or induce resistance mutations at a low level is the most promising direction. Therefore, the research presented by the authors is innovative, and further exploration of the topic by studying the mechanisms of action, efficacy, and in vivo toxicity of the selected compounds may lead to the development of effective antibacterial therapies.
The research methodology used is adequate for the research problem addressed (determination of minimum inhibitory concentrations MIC, fractional inhibitory concentration FIC values for both monotherapy and combination therapy, evaluation of resistance over evolutionary cycles). The methods are clearly and comprehensibly described, and in my opinion, they can be replicated by other researchers. The results obtained are presented in tables and graphs, and their analysis is described in an interesting way. However, in my opinion, the data should be supplemented with MIC values for reference compounds for both monotherapy and combination therapy. There is a lack of a general structure-activity relationship characterization for the studied compounds. Although the selected compounds are chemically diverse, it can be observed that fluconazole was the least active compound in both monotherapy and combination therapy. Only this one compound belongs to the 1,2,3-triazoles, while the other three belong to the imidazoles. The reviewer questions this selection of compounds. The group of tested compounds could be expanded, for example, by including molecules containing other azole groups. The article could benefit from including structural formulas of the studied compounds. Although this aspect is often omitted in biological studies, it could substantively enrich the article. An editorial error is present in line 397, where a space is missing before the literature reference [56].
The article is written in a logical and coherent manner, with a division into introduction, methods, results, discussion, and conclusions. The writing style is precise and understandable for the reader. The article refers to current and relevant literature in the field of azoles exhibiting biological activity. Previous studies described in articles [25, 55, 56] provide a good introduction to the research in the reviewed article. The literature references [20] and [30] should be supplemented with the missing information, including the DOI number. In the reviewer's opinion, the article, after minor revisions should be accepted for publication.
Author Response
We have put the reply in the attachment, thank you kindly for reviewing it.

Reviewer 3 Report
Comments and Suggestions for Authors
The author reported in this manuscript “Imidazole’s and Quaternary Ammonium Compounds as Effective Therapies Against (Multidrug-Resistant) Bacterial Wound Infections” the world facing most important and critical issue antimicrobial resistance in wound pathogens by investigating the antibacterial activity of imidazole’s combined with quaternary ammonium compounds (QACs). The novelty of this work lies in repurposing azoles, which are traditionally used as antifungals, for antibacterial applications, a concept that is innovative and valuable. The rationale is well-supported by previous studies demonstrating synergistic antifungal activity of azoles and QACs, and the potential to extend this synergy to bacterial pathogens. The use of checkerboard assays to identify synergistic combinations is appropriate, and the focus on multidrug-resistant gram-positive pathogens is important, given their relevance in wound infections. However, the manuscript could benefit from more detailed descriptions of the concentrations used in the assays and any limitations encountered. The identification of several effective combinations, particularly the miconazole and domiphen bromide pairing, is a highlight of the study. This combination exhibits broad-spectrum activity and does not induce resistance over 11 bacterial evolution cycles. Despite these promising findings, the discussion could be strengthened by offering insights into the potential mechanisms of action for these combinations. Speculation on membrane disruption or enzyme inhibition, for example, would provide context for future research. Additionally, while the study focuses on gram-positive pathogens, exploring the activity of these combinations against gram-negative bacteria would broaden the scope and impact of the findings. Overall, the manuscript correctly emphasizes the need for further research into the in vivo efficacy and toxicity of these combinations, particularly for topical wound treatments. Future studies should also explore pharmacokinetics and pharmacodynamics to facilitate translation into clinical applications. I am recommending for the publication of this article after the following recommended changes in the manuscript:
1. Did the author investigated or hypothesized the specific mechanisms by which imidazole’s and quaternary ammonium compounds (QACs) exhibit synergistic activity against bacterial pathogens? Could it be possible to work these combinations disrupt the bacterial cell membrane or target other cellular components?
2. Did author observe no resistance development after 11 cycles of evolution with the miconazole/domiphen bromide combination. How does the explain this stability, and does the plan to extend these evolution studies to test for resistance over longer periods?
3. Could the author provide more detailed information on the concentrations of both imidazole’s and QACs used in the checkerboard assays? How does these concentrations compare to clinically relevant dosages for topical applications?
4. Although the study focuses on gram-positive pathogens, have the author considered testing these combinations against gram-negative bacteria? Given the rising concern over multidrug resistance in gram-negative organisms, this could be an important extension of the author’s research.
5. Has the author conducted preliminary toxicity assessments of these combinations in cell cultures or animal models? What are your plans for in vivo efficacy studies to evaluate their potential as topical wound treatments?
6. Besides wound infections, could these synergistic combinations be considered for other types of infections (e.g., skin or mucosal infections)? Does it anticipate broader clinical applications beyond wound care?
Author Response

(The authors gave the same response as above.)

Reviewer 4 Report
Comments and Suggestions for Authors
The document is adequate and interesting, however, the following corrections and suggestions are recommended to improve the work.
At the level of the entire document, correct the wording in an impersonal manner, particularly avoid the use of "we"
Correct the use of italics throughout the document for scientific names
Line 12 in the abstract there is an underscore at the beginning that is not correct
Lines 17 and 19, write in an impersonal manner
Line 39 correct micro-organisms for microorganisms
Line 56, the text contains a lot of information for a single reference, regarding 17, 18 and 19
Line 94, indicate which sigma code corresponds to miconazole nitrate salt, as well as the other drugs used, this in order to guarantee the reproducibility of the studies
Line 102 define the use of dH2O
Line 159 include more references of the mathematical function used, reference 39 is superficial to the calculation, so other references of the analysis should be included
Line 205 what distribution does the normal data present or not normal, the type of statistical test used needs to be justified
Line 324, in figure 1, was the MIC value presented average? If so, the error bar needs to be included to determine the reliability of the data analysis, in section a, which involves the fact that in cycle 8 the MONO MICO decreased, this is not discussed in the text
It is recommended that the conclusions be rewritten, the text is correct but complicated to follow and to include the results.
Comments on the Quality of English Language
The document needs a language review to reduce long sentences and avoid writing in personal terms
Author Response
Please see the attachment for our reply, thank you kindly for reviewing our work.
